# A universal strategy towards high-energy aqueous multivalent-ion batteries

Xiao Tang[1], Dong Zhou [1✉], Bao Zhang [2,3], Shijian Wang[1], Peng Li[4], Hao Liu [1], Xin Guo[1], Pauline Jaumaux[1], Xiaochun Gao[1], Yongzhu Fu [5], Chengyin Wang[6], Chunsheng Wang [2✉] & Guoxiu Wang [1✉]

Rechargeable multivalent metal (*e.g.*, Ca, Mg or, Al) batteries are ideal candidates for large-scale electrochemical energy storage due to their intrinsic low cost. However, their practical application is hampered by the low electrochemical reversibility, dendrite growth at the metal anodes, sluggish multivalent-ion kinetics in metal oxide cathodes and, poor electrode compatibility with non-aqueous organic-based electrolytes. To circumvent these issues, here we report various aqueous multivalent-ion batteries comprising of concentrated aqueous gel electrolytes, sulfur-containing anodes and, high-voltage metal oxide cathodes as alternative systems to the non-aqueous multivalent metal batteries. This rationally designed aqueous battery chemistry enables satisfactory specific energy, favorable reversibility and improved safety. As a demonstration model, we report a room-temperature calcium-ion/sulfur||metal oxide full cell with a specific energy of 110 Wh kg$^{-1}$ and remarkable cycling stability. Molecular dynamics modeling and experimental investigations reveal that the side reactions could be significantly restrained through the suppressed water activity and formation of a protective inorganic solid electrolyte interphase. The unique redox chemistry of the multivalent-ion system is also demonstrated for aqueous magnesium-ion/sulfur||metal oxide and aluminum-ion/sulfur||metal oxide full cells.

[1] Centre for Clean Energy Technology, Faculty of Science, University of Technology Sydney, Sydney, NSW, Australia. [2] Department of Chemical and Biomolecular Engineering, University of Maryland, College Park, MD, USA. [3] School of Optical and Electronic Information, Huazhong University of Science and Technology, Wuhan, PR China. [4] College of Material Science and Engineering, Nanjing University of Aeronautics and Astronautics, Nanjing, PR China. [5] College of Chemistry and Molecular Engineering, Zhengzhou University, Zhengzhou, China. [6] College of Chemistry and Chemical Engineering, Yangzhou University, Yangzhou, China. ✉email: zhoudong087@gmail.com; Cswang@umd.edu; Guoxiu.Wang@uts.edu.au

Reliable large-scale energy storage is indispensable for integrating renewable energies (*e.g.* solar and wind) into electric grids[1]. As cost-effective alternatives to lithium (Li)–ion batteries, rechargeable multivalent–ion batteries (MIBs) are ideal energy storage technologies for grid-scale applications[2]. Among many multivalent cations, $Ca^{2+}$, $Mg^{2+}$, and $Al^{3+}$ are of particular interest owing to their non–toxicity, stable valence states, relatively small ionic radii, low redox potentials ($Ca/Ca^{2+}$: –2.87 V; $Mg/Mg^{2+}$: –2.36 V; $Al/Al^{3+}$: –1.68 V vs. standard hydrogen electrode (SHE)), and natural abundance (Ca: 4.86 wt%; Mg: 2.60 wt%; Al: 8.21 wt% in earth crust, whereas Li is only 0.0065 wt%)[3,4]. More importantly, such multivalent cations can transfer more than one electron per cation, which can increase specific energies[5].

Despite the above merits, the research progress on MIBs is far from satisfactory. The first challenge is the slow kinetics of multivalent cation insertion/extraction in metal oxide cathodes in organic-based electrolytes. Metal oxides[6] generally exhibit higher potential and/or capacity than other cathode materials (e.g., sulfides[7], polyanions[8], Prussian blue analogs[9], and organic compounds[10]), which benefits high-specific energy. However, the strong electrostatic interactions between multivalent ions and organic solvent molecules as well as cathode host lattices leads to sluggish cation diffusion, which triggers huge polarization and poor cycling stability[11]. Furthermore, the conversion reactions also plague the cathode materials in multivalent–ion batteries[12]. Regarding anodes, multivalent cations are difficult to penetrate through the organic component-rich passivating interphase layers on the surfaces of Ca, Mg, and Al metal anodes[13,14], in which the organic components are mainly formed due to the reduction of organic solvents in electrolytes[15]. Only few organic-based electrolytes using highly flammable ether solvents (e.g., tetrahydrofuran[16]) can avoid the formation of organic component-rich passivating interphase layers and thus support the reversible plating/stripping of multivalent metal anodes. However, the uncontrollable dendrite growth on the multivalent metal surfaces causes huge safety concern[17]. Furthermore, the low oxidation resistance (<0.9 V vs. SHE) of such ether-based electrolytes inhibits the use of high-voltage metal oxide cathodes[18]. Other anode materials such as alloys[19], carbonaceous materials[20], and polymeric anodes[21] can only deliver limited capacities (generally <300 mAh g⁻¹) in MIBs. Therefore, it is essential to re-design battery chemistry for the development of MIBs.

To re-invent the MIB chemistry by rationally designing high-voltage aqueous batteries, highly concentrated aqueous gel electrolytes featured with low toxicity and non–flammability should be used to replace flammable organic ether-based electrolytes. Such aqueous gel electrolytes have vastly expanded the voltage windows of aqueous electrolytes, and thereby support high-specific energy electrochemical redox couples based on high-voltage metal oxide cathodes ($M_xMnO_2$, M represents Ca, Mg, and Al). Moreover, the kinetics of multivalent cation insertion/extraction in metal oxide cathodes is fast in aqueous gel electrolytes due to water/proton co–insertion[22]. We chose sulfur as multivalent–ion conversion anode material to avoid the irreversible plating/stripping and dendrite growth of multivalent metal anodes. The high theoretical capacity of sulfur (1675 mAh g⁻¹) is close to those of multivalent metal anodes (Ca, Mg, and Al)[23]. More importantly, the relatively high potential of sulfur in aqueous electrolytes can avoid the formation of organic component-rich interphase layers originated from the reduction of organic solvent, which have been validated in aqueous Li-ion/sulfur batteries[24,25]. Conversely, in the highly concentrated aqueous gel electrolytes, a protective inorganic solid electrolyte interphase (SEI) formed on sulfur anode enables a highly reversible anodic polysulfide conversion, which is usually problematic in

non–aqueous MIBs[26]. Furthermore, the high-voltage metal oxide cathodes endow the aqueous full cells with high output voltages comparable to those of non–aqueous MIBs, which ensures to achieve high-specific energies. The aqueous multivalent–ion/sulfur||metal oxide system can effectively circumvent the drawbacks of the non–aqueous multivalent battery chemistry, thus leading towards efficient energy and power performances as well as improved safety aspects; it should be noted that aqueous multivalent–ion/sulfur||metal oxide batteries are not an extension of aqueous Li-ion/sulfur||metal oxide batteries, which increases the battery safety by scarifying the specific energy[24].

Here, we show, aqueous multivalent–ion/sulfur| |metal oxide chemistry could be successfully deployed in the most challenging Ca-based battery system. The as-developed full aqueous Ca–ion/sulfur battery (ACSB) consists of a sulfur/carbon (S/C) anode, a layered $Ca_{0.4}MnO_2$ cathode, and a gel electrolyte based on 8.37 m (mol kg⁻¹ₛₒₗᵥₑₙₜ) saturated $Ca(NO_3)_2$ aqueous solution. The low-water activity of gel electrolyte effectively suppresses the capacity loss caused by calcium polysulfide dissolution in the anode, and simultaneously facilitates a fast and stable calcium ion intercalation/deintercalation in the cathode. The as-developed ACSBs deliver a high-specific energy of 110 Wh kg⁻¹, 83% capacity retention after 150 cycles at 0.2 C, and superior safety in the aqueous gel electrolyte. Furthermore, we demonstrate that this strategy can also be extended for building high-energy aqueous Mg–ion/sulfur||metal oxide and Al–ion/sulfur||metal oxide batteries.

## Results and discussion

**Highly concentrated aqueous gel electrolyte.** $Ca(NO_3)_2$ salt was chosen for preparing electrolytes because it has a high solubility (up to 8.37 m) in water comparable to those of fluorinated Ca salts (calcium (II) bis(trifluoromethylsulphonyl)imide (Ca(TFSI)₂) and calcium (II) trifluoromethanesulfonate (Ca(OTf)₂), etc.). Meanwhile, its low cost and eco-friendliness benefit for large-scale applications. It is well known that the electrochemical stability window, imposed by hydrogen evolution reaction (HER) on the anode and oxygen evolution reaction (OER) on the cathode, restricts the application of electrode materials in aqueous batteries. The electrochemical stability window should be wide enough to support the redox couples of the battery chemistry, because even trace amounts of hydrogen or oxygen evolution will seriously deteriorate the cycle life and Coulombic efficiency of batteries[27]. We evaluated the electrochemical stability windows of the aqueous $Ca(NO_3)_2$ electrolytes with different molalities and a gel electrolyte with 10 wt% polyvinyl alcohol (PVA) dissolved in saturated $Ca(NO_3)_2$ aqueous solution by linear sweep voltammetry (LSV) on stainless-steel electrodes. The 1 m $Ca(NO_3)_2$ aqueous electrolyte exhibits weak acidity (Supplementary Fig. 1), and delivers a narrow stability window of only ≈1.95 V (Fig. 1a). The onset potential for HER in electrolytes continually decreases with the increase of $Ca(NO_3)_2$ concentration from –0.8 V vs. Ag/AgCl in the 1 m $Ca(NO_3)_2$ electrolyte to –1.1 V in the saturated electrolyte, and further decreases to –1.2 V after adding of PVA, which is far beyond the thermodynamic stability limitation of water (the left panel of Fig. 1a). Such negative shifts of HER could be attributed to the reduction of free water molecules and the protection of SEI on the anode, which ensures a reversible calciation/de-calciation of sulfur to be within the voltage window of the aqueous gel electrolyte (Fig. 1b). Meanwhile, on the cathode side, the onset potential for OER also increases from 1.15 V vs. Ag/AgCl in the 1 m $Ca(NO_3)_2$ electrolyte to 1.4 V in the aqueous gel electrolyte (the right panel of Fig. 1a), mainly owing to the suppressed water activity together with an inner Helmholtz layer populated by $NO_3^-$ anions[25]. This envelops the redox potentials

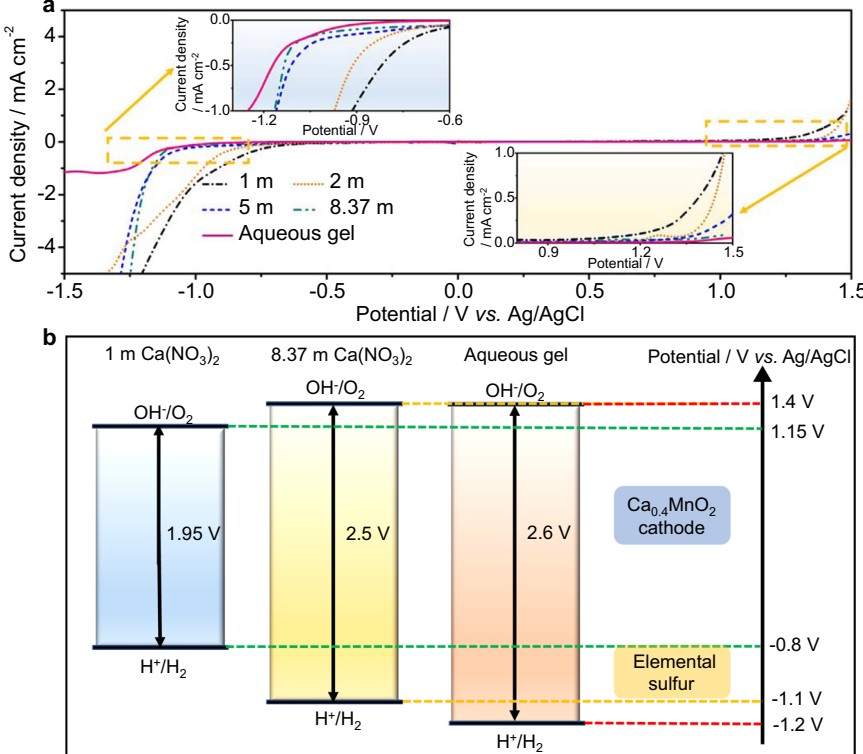

**Fig. 1 The electrochemical stability window of electrolytes. a** Linear voltammetry curves recorded at 1 mV s$^{-1}$ in 1 m, 2 m, 5 m, saturated (8.37 m) Ca(NO$_3$)$_2$ electrolytes and aqueous gel electrolyte. The insets are the magnified views of the regions marked near anodic and cathodic extremes. **b** The electrochemical stability windows of electrolytes, and the redox voltages of Ca$_{0.4}$MnO$_2$ cathode and sulfur anode obtained from experimental data.

of the Ca$_{0.4}$MnO$_2$ cathode (≈0.2 V vs. Ag/AgCl, Fig. 1b). Overall, this result demonstrates that the synergistic effect of high salt concentration and PVA successfully expands the electrochemical stability window to 2.6 V to fulfill the electrochemical redox couple of sulfur anode and Ca$_{0.4}$MnO$_2$ cathode.

The aqueous electrolytes were studied by both molecular dynamics (MD) simulations and experimental investigations. The MD simulations were employed to study the structure evolution of diluted/saturated Ca(NO$_3$)$_2$ solution and aqueous gel electrolyte (Fig. 2a–c). In 1 m Ca(NO$_3$)$_2$ solution with a salt-to-water molar ratio (S:W, i.e., Ca(NO$_3$)$_2$:H$_2$O) of ≈1:55, individual Ca$^{2+}$ ions are observed to evenly distributed in the solvent, and on average 6.2 water molecules are coordinated with one Ca$^{2+}$ to form a primary solvation sheath (Fig. 2a). Noticeably, only ≈10.9% of water molecules are coordinated with Ca$^{2+}$ ions, while others interact with each other through hydrogen bonds (Fig. 2d and Supplementary Fig. 2). Such huge amount of free water molecules usually triggers preferential hydrogen evolution, and prevents the reaction between Ca$^{2+}$ and sulfur. Meanwhile, the NO$_3^-$ are scattered randomly among the water molecules without any coordination with Ca$^{2+}$ cations. When the concentration increases to 8.37 m with a S:W of ≈1:6.6, large amount of Ca$^{2+}$ ions tend to partially share the primary water sheaths with each other, and more water molecules (63.1%, Fig. 2d and Supplementary Fig. 2) are coordinated with Ca$^{2+}$, which significantly reduces the activity of water molecules[25]. Moreover, on average three NO$_3^-$ anions are observed in each Ca$^{2+}$ primary solvation sheath (Fig. 2b), resulting in an interfacial chemistry dominated by the NO$_3^-$ reduction. After dissolving PVA into the saturated solution, the Ca$^{2+}$–H$_2$O complexes exhibit a polymer-like aggregation (Fig. 2c). This suggests that large amount of water molecules is immobilized by highly concentrated Ca(NO$_3$)$_2$ salt and polymer chains. Furthermore, as shown in Fig. 2d and

Supplementary Fig. 3, the hydrogen bonds of diluted Ca(NO$_3$)$_2$ solution are ≈1.35 per water molecule, while this value decreases to ≈1.20 with the concentration increasing to 8.37 m. This suggests a reduction of hydrogen bonds at the saturated state due to the high fraction of coordinated water number. In contrast, hydrogen bonds increase to ≈1.25 in the gel electrolyte, which is mainly due to the formation of hydrogen bonds between the hydroxy group in the PVA and water molecules in the Ca$^{2+}$–H$_2$O complexes (Fig. 2d). Such perturbation of the water hydrogen bond network by water–PVA interactions can further reduce the activity of water solvent (see the distance change between two nearest water molecules shown in Supplementary Fig. 4)[28], thus effectively enhancing the electrochemical stability and suppressing the diffusion of the polysulfides into water.

Noticeably, the aqueous gel electrolyte shows an ionic conductivity of 11.56 mS cm$^{-1}$ at 25 °C (Supplementary Fig. 1), which is higher than those of most organic-based electrolytes[29]. Raman spectroscopy measurements were carried out to investigate the interplay among ions, water, and PVA. Two peaks at ≈717 and 740 cm$^{-1}$ emerge in the aqueous electrolytes with Ca(NO$_3$)$_2$ concentration higher than 5 m (Fig. 2e, left panel), suggesting the formation of ion pairs (see crystalline Ca(NO$_3$)$_2$·4H$_2$O as the reference in Supplementary Fig. 5)[30]. The O–H stretching vibration modes of water molecules in aqueous electrolytes are presented in the right panel of Fig. 2e. In the 1 m and 2 m dilute Ca(NO$_3$)$_2$ electrolytes, the O–H stretching vibrations exhibit broad Raman bands consisting of several components, which are attributed to water molecules with different hydrogen–bonding environments in water clusters[31]. Both the width and intensity of this band shrink with the increase of the electrolyte concentration, indicating that the Ca$^{2+}$–H$_2$O coordination will break the water's hydrogen–bonding structuring. This is well consistent with the red shifts in the $^1$H Nuclear

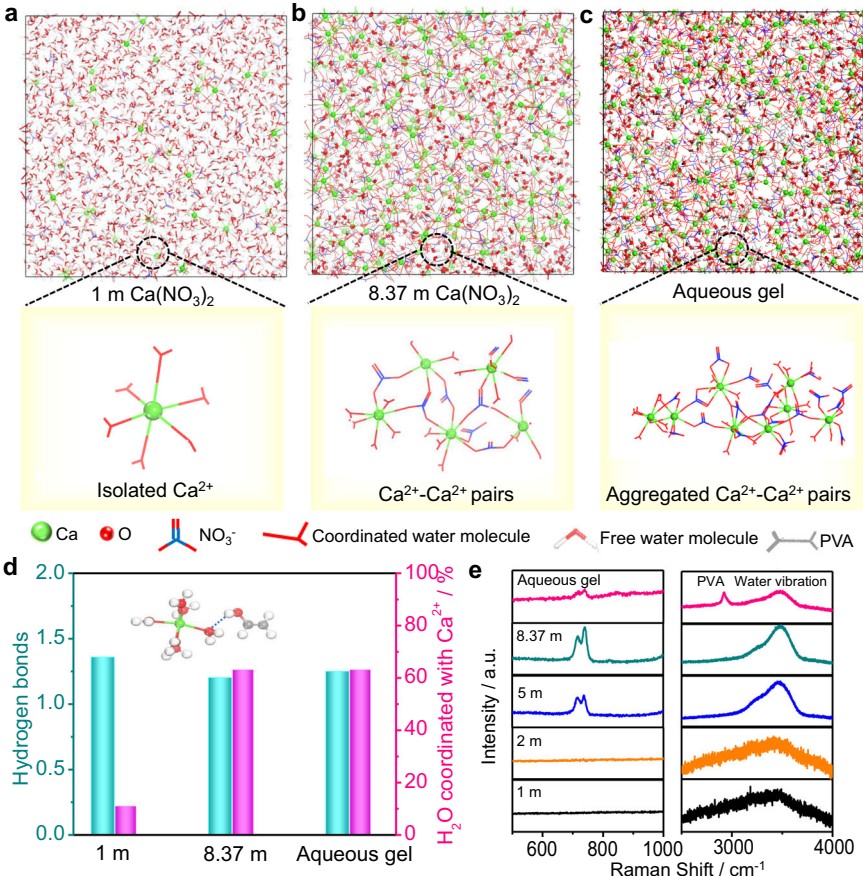

**Fig. 2 Molecular dynamics simulations and characterization of electrolytes.** Snapshots of local structure evolution for **a** 1 m Ca(NO$_3$)$_2$ electrolyte, **b** saturated Ca(NO$_3$)$_2$ electrolyte, and **c** aqueous gel electrolyte based on MD simulation at 10 ns. **d** The hydrogen bonds and the percentage of water molecular coordinated with Ca$^{2+}$ for three electrolyte samples based on MD simulation at 10 ns. The hydrogen bond between the Ca$^{2+}$–H$_2$O complex and PVA repetitive unit is shown in the inset. The green, red, white, and gray balls represent Ca, O, H, and C, respectively. **e** Raman spectra of the 1 m, 2 m, 5 m and saturated Ca(NO$_3$)$_2$ aqueous electrolytes, and aqueous gel electrolyte.

magnetic resonance (NMR) spectra (Supplementary Fig. 6) and the density functional theory (DFT) calculation results (Supplementary Fig. 7). The aqueous gel electrolyte showed only a small hump at ≈3500 cm$^{-1}$ in the Raman spectrum, indicating that the water clusters were significantly diminished in this quasi–solid–state electrolyte.

**Conversion mechanism and performances of sulfur/carbon composite anode.** Although the solubility of elemental sulfur is negligible in the aqueous electrolytes (Supplementary Fig. 8), the short-chain polysulfides (S$^{2-}$–S$_4^{2-}$) are highly soluble in water[32], which subsequently shuttle between the electrodes and trigger active material loss and interfacial deterioration. Therefore, to improve the electrochemical performances of ACSBs, it is crucial to decrease the solubility of polysulfides in aqueous electrolyte. It is known that a high solubility of polysulfides will increase the concentration gradient between the sulfur electrode|electrolyte interface and the electrolyte bulk phase, thus accelerating the polysulfides' diffusion according to Fick's Law[33]. Therefore, the solubility of polysulfides can be evaluated based on their calculated diffusivity. MD simulations were performed to investigate the diffusion of a typical soluble polysulfide, CaS$_4$, in different electrolytes (Fig. 3a, Supplementary Fig. 9, and Supplementary Videos 1–3). As shown in left panel of Fig. 3a, CaS$_4$ diffuses quickly in diluted solution within a short time (10 ns), while the diffusion can be significantly suppressed by applying higher salt

concentration and PVA chains. Indeed, although the soluble polysulfides can extract water from electrolyte to form anolyte until reaching equilibrium[24], the MD simulations demonstrate the severe aggregation of calcium polysulfides with domain formation in aqueous gel electrolyte. The as-formed mixture of solid and liquefied anolyte is expected to be not miscible with the bulk gel electrolyte. To experimentally verify this, CaS$_4$ aqueous solution with dark brown color was added onto the surfaces of electrolyte samples. As shown in Fig. 3b, the added CaS$_4$ immediately diffused into the 1 m Ca(NO$_3$)$_2$ diluted electrolyte, whose color turned to yellow in seconds. The saturated Ca(NO$_3$)$_2$ electrolyte, however, presented a clear segregation interface between the added CaS$_4$ solution for up to 2 days. Meanwhile, yellow anolyte can be observed to diffuse below the interface, and the segregation interface became blurry after aging for 5 days. In sharp contrast, the aqueous gel electrolyte remained a clear segregation interface after aging for 5 days. These experiments support the claims that the dissolution of calcium polysulfides is negligible in the gel electrolyte, which is attributed to the suppressed water activity by the high salt concentration and PVA.

A sulfur/carbon composite was applied as anode material for the ACSBs, since the porous carbon (Supplementary Fig. 10) can enhance the electronic conductivity of sulfur and further inhibit the dissolution of calcium polysulfides into electrolyte[23]. The chemically stable titanium/stainless-steel meshes were used as current collectors in ACSBs. The mesh current collector not only can reduce the strain on the electrodes during cycling to avoid the

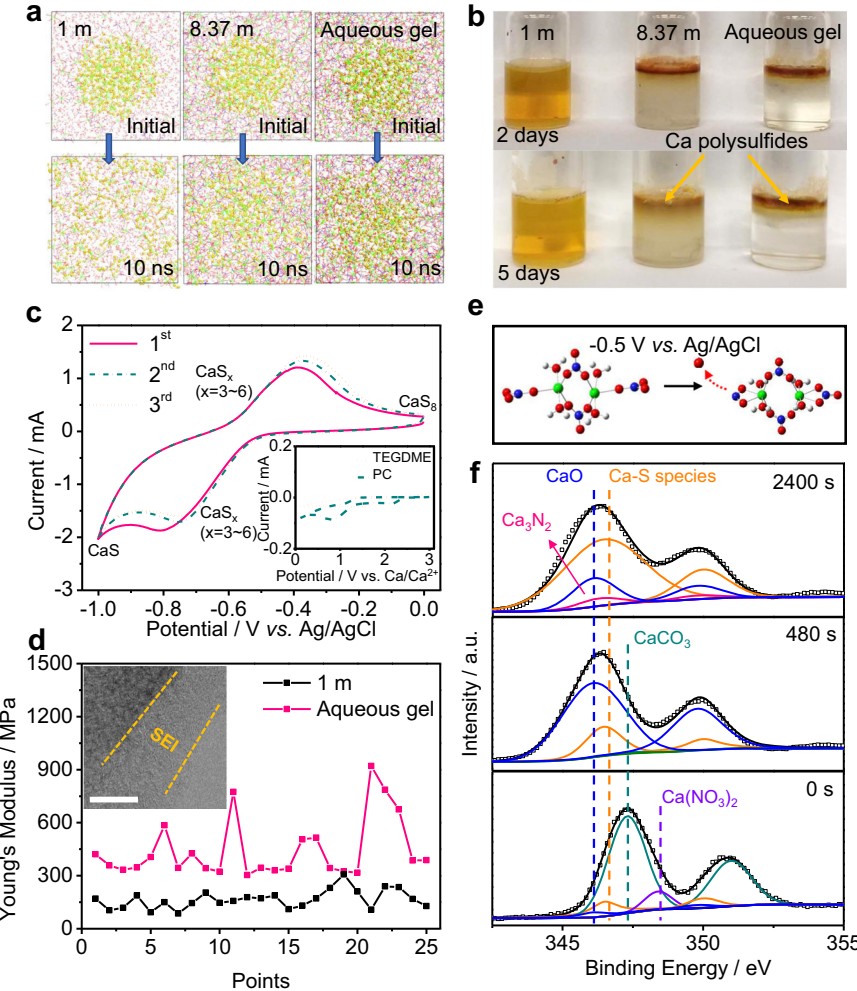

**Fig. 3 Reaction mechanism of elemental sulfur in the aqueous gel electrolyte. a** Local structure evolutions of $CaS_4$ (yellow) diffusions in 1 m, 8.37 m Ca $(NO_3)_2$ electrolytes, and aqueous gel electrolyte based on MD simulation. The yellow balls represent S atoms, and other symbols are same with those in Fig. 2a–c. **b** Visual observation of calcium polysulfides diffusion in 1 m, 8.37 m $Ca(NO_3)_2$ electrolyte, and gel electrolyte. **c** The initial CV curves of S/C electrode collected at a scan rate of 0.2 mV s$^{-1}$ in the aqueous gel electrolyte. The CV curves of Ca metal||S/C cell with 0.5 m $Ca(OTf)_2$ in TEGDME or 0.5 m $Ca(OTf)_2$ in PC electrolytes at 0.2 mV s$^{-1}$ are shown in inset of Fig. 3c. **d** The Young's Modulus of S/C anodes tested in 1 m $Ca(NO_3)_2$ and aqueous gel electrolytes. The points are selected from the AFM scanning images in Supplementary Fig. 16. The inset is HADDF–STEM images of S/C anodes after cycling in aqueous gel electrolyte, scale bar: 10 nm. **e** The DFT calculations of reduction of $Ca^{2+}(NO_3^-)_3(H_2O)_x$ aggregate. The green, blue, red, and white balls represent Ca, N, O, and H atoms, respectively. **f** In-depth Ca 2p XPS of the S/C electrode after cathodic CV scanning.

formation of cracks, but also provides three–dimensional charge transport path and opening spaces to accommodate active materials[34]. Cyclic voltammetry (CV) was carried out to explore the reaction mechanism of S/C anode in different electrolytes. It is seen that in non–aqueous electrolytes based on organic solvents (such as propylene carbonate (PC) and tetraethylene glycoldimethyl ether (TEGDME)), the CV curves show cathodic peaks at 1.5–0.75 V vs. $Ca/Ca^{2+}$, while almost no peaks are observed in the anodic scan (inset of Fig. 3c). This indicates an irreversible sulfur conversion chemistry in non–aqueous Ca–S batteries. In contrast, in the aqueous gel electrolyte, a gentle current slope starts at ≈0 V vs. Ag/AgCl during the initial cathodic scan, suggesting the transformation from sulfur to $CaS_x$ (x = 4–8) species. This is consistent with the ex situ Raman spectrum in Supplementary Fig. 11 (see the $S_8$ (≈151 cm$^{-1}$) and $S_4^{2-}$–$S_8^{2-}$ (≈746 cm$^{-1}$) peaks[23], black curve). Then, a redox peak at −0.55 to −0.90 V is observed, which can be assigned to the solid-liquid transition from long-chain polysulfides to short-chain polysulfides such as $CaS_4$ (see the intensity decreases of the $S_8$ and $S_{4–8}^{2-}$ peaks together with the appearance of $S_4^{2-}$ peak at ≈255 cm$^{-1}$ in

the Raman spectrum[35], red curve in Supplementary Fig. 11). Meanwhile, CaS and trace amount of $Ca(HS)_2$ can be detected as the final discharge products based on the in-depth X-ray photoelectron spectroscopy (XPS, Supplementary Fig. 12) and ultraviolet–visible spectroscopy (UV–vis, Supplementary Fig. 13) results. The CaS and $Ca(HS)_2$ are generated from the reductions of solid calcium polysulfide aggregates (anhydrous) and liquefied calcium polysulfide anolytes (hydrous), respectively. For the anodic scan, a gentle hump from −1 to −0.6 V is related to the oxidation of short-chain polysulfides to $S_5^{2-}$ (415 and 495 cm$^{-1}$ in the Raman spectrum) and $SO_3^{2-}$ (612 cm$^{-1}$ in the Raman spectrum, see the orange curve in Supplementary Fig. 11). Then, one peak at around −0.6 to −0.2 V appears, indicating that the majority of sulfur species have been transformed to elemental sulfur (see the cyan curve in Supplementary Figs. 11 and 14). Therefore, the above results clearly elucidate a reversible sulfur conversion chemistry in the highly concentrated aqueous gel electrolyte.

The interfacial properties, including morphology and strength of the SEI layers, have been investigated by high–angle annular

dark field (HAADF)–scanning transmission electron microscope (STEM) and atomic force microscope (AFM). As shown in the STEM image, a $\approx$ 3 nm SEI was constructed on the S/C electrode surface after a cathodic CV scanning in 8.37 m $Ca(NO_3)_2$ saturated electrolyte (Supplementary Fig. 15b), meanwhile no SEI was observed in the 1 m $Ca(NO_3)_2$ dilute electrolyte (Supplementary Fig. 15a). The SEI layer in aqueous gel electrolyte exhibited an amorphous structure with a thickness of $\approx$10 nm due to the participation of PVA polymer matrix (inset of Fig. 3d and Supplementary Fig. 15c). In addition, the surface of SEI layer–coated S/C anode cycled in the aqueous gel electrolyte shows an average Young's modulus of $\approx$445 MPa, which is much higher than that cycled in the 1 m $Ca(NO_3)_2$ dilute electrolyte ($\approx$165 MPa, Fig. 3d and Supplementary Fig. 16). Such robustness of SEI is expected to maintain its structural integrity against the electrode deformation during cycling.

The interfacial chemistry occurring on the surface of S/C electrode is further revealed through density functional theory (DFT) calculations and in-depth XPS measurement. As discussed above, $NO_3^-$ ions are not coordinated with $Ca^{2+}$ cations in the diluted electrolytes, while on average three $NO_3^-$ anions are coordinated with one $Ca^{2+}$ solvation sheath in the aqueous gel electrolyte (Fig. 2c). Consequently, the reduction potential of $NO_3^-$ is greatly altered by its intimate interaction with $Ca^{2+}$. Based on the DFT calculation results shown in Fig. 3e, the $Ca^{2+}$ $(NO_3^-)_3(H_2O)_x$ decomposes below –0.5 V versus Ag/AgCl, which is significantly higher than the reduction potential of an isolated $NO_3^-$ anion at –0.84 V (Supplementary Fig. 17) and the potential of HER. Therefore, the preferential reduction of $NO_3^-$ anions facilitates the formation of SEI layer on the S/C electrode surface to suppress the further anion decomposition. The Ca 2p in-depth XPS spectra of the as-formed SEI on S/C electrode are presented in Fig. 3f. In Ca $2p_{3/2}$ doublet, the five peaks at 346.1, 346.3, 346.5, 347.3, and 348.4 eV can be assigned to the CaO, $Ca_3N_2$, Ca–S species (CaS, $Ca(HS)_2$, etc.), $CaCO_3$, and $Ca(NO_3)_2$, respectively[36–40]. The peak intensity of $CaCO_3$ decreases, while the intensities of CaO, $Ca_3N_2$, and Ca–S species increase with increasing sputtering time. This demonstrates that the $CaCO_3$ (formed from the trace dissolved $CO_2$) mainly distributes in the SEI outer layer, while the CaO and $Ca_3N_2$ (formed from the decomposition of $NO_3^-$) is the main components of the inner layer of the as-formed SEI[9]. Noticeably, these SEI components can only stably exist as solid deposits on the electrode surface in highly concentrated electrolytes, because they would hydrolyze and dissolve quickly in the water media. Such an inorganic SEI is expected to benefit a negative shift of the HER and a further inhibition of the polysulfide dissolution[25].

**Structure evolution and performances of $Ca_{0.4}MnO_2$ cathode.**
The cathode material of the ACSBs, $Ca_{0.4}MnO_2$, was synthesized via a facile in situ electrochemical transformation from $Mn_3O_4$ precursor. The $Mn_3O_4$ working electrode was assembled into a beaker cell with the saturated $Ca(NO_3)_2$ electrolyte, platinum (Pt) counter electrode, and Ag/AgCl reference electrode. Then, the cell was cycled in the voltage window of –0.5 to 1 V vs. Ag/AgCl for electrochemical conversion. During the electrochemical oxidation process, $Mn^{2+}$ in the spinel $Mn_3O_4$ continuously dissolved into the electrolyte accompanying with the oxidation of $Mn^{3+}$ to $Mn^{4+}$, which resulted in a structure evolution to form layered $MnO_2$ with rearrangement of Mn[41,42]. $Ca^{2+}$ cations can be intercalated into the $MnO_2$, thus forming the birnessite $Ca_{0.4}MnO_2$ (the molecular formula is determined based on an overall Ca:Mn atomic ratio of $\approx$0.4 according to the inductively coupled plasma (ICP) measurement, Supplementary Table 1) in the subsequent reduction process (Fig. 4a). The STEM analysis

was conducted to characterize the cathode material structure. As shown in Fig. 4b and Supplementary Fig. 18a, the $Mn_3O_4$ particles with an average diameter of $\approx$50 nm shows a homogeneous structure with typical atom projection arrangement of spinel. Meanwhile, the selected–area electron diffraction (SAED) pattern (the inset in Fig. 4b) illustrates a tetragonal structure in $Mn_3O_4$ precursor[43]. After in situ electrochemical transformation, the $Ca_{0.4}MnO_2$ nanoparticles become porous with blunted edges (Supplementary Fig. 18b). Moreover, a layered structure with an interlayer spacing of $\approx$6 Å can be observed in the corresponding HAADF–STEM image and SAED pattern (Fig. 4c). The elemental mapping of $Ca_{0.4}MnO_2$ in Supplementary Fig. 19 confirms the $Ca^{2+}$ intercalation in the $Ca_{0.4}MnO_2$. Synchrotron X-ray powder diffraction was performed to further identify the phase and composition of the cathode material. As shown in Fig. 4d and Supplementary Fig. 20, the $Mn_3O_4$ precursor exhibits a typical spinel structure with space group of I41/amd[44]. After electrochemical oxidation, the $MnO_2$ as charge product presents peaks at $\approx$6.8° and 20° with most of other sharp peaks vanish, which indicates the phase transformation from spinel to birnessite[45]. These newly formed peaks shift towards lower angles in the pattern of $Ca_{0.4}MnO_2$ after the reduction process, demonstrating that intercalation of $Ca^{2+}$ induces variations in the lattice parameter in the layered $MnO_2$ structure. Moreover, as shown in Mn 3 s XPS (Supplementary Fig. 21), the energy separations ($\Delta E$) were measured to be 5.64, 4.55, and 5.10 eV for the $Mn_3O_4$, birnessite $MnO_2$, and $Ca_{0.4}MnO_2$ samples, respectively. According to the linear relationship between the chemical valence of Mn and the $\Delta E$ value, the average oxidation states of Mn were calculated to be $\approx$2.5, 3.95, and 3.19 for $Mn_3O_4$, birnessite $MnO_2$, and $Ca_{0.4}MnO_2$ samples, respectively[46,47]. This verifies the electrochemical transformation mechanism to form the $Ca_{0.4}MnO_2$.

Electrochemical measurements were conducted to characterize the performances of the $Ca_{0.4}MnO_2$ cathode. The CV curves in Fig. 4e show a pair of non–sharp redox peaks located at $\approx$0.2 V, which can be attributed to the reversible intercalation/deintercalation of $Ca^{2+}$ in the layered structure[48]. Based on Dunn's method[49], it can be speculated that the capacity at peak region is dominated by a diffusion-controlled process (Supplementary Fig. 22). The initial CV curves are highly overlapped, indicating that the $Ca_{0.4}MnO_2$ is highly reversible at room temperature. The $Ca_{0.4}MnO_2$ cathode can deliver high capacities of 210, 170, 135, 116 mAh $g^{-1}$ based on the mass of $Ca_{0.4}MnO_2$ at specific currents of 10, 50, 100, and 200 mA $g^{-1}$ (Supplementary Fig. 23a), which are higher than those of the previously reported cathode materials for Ca-based batteries[21]. It should be noted that the $Ca_{0.4}MnO_2$ cathode is non–rechargeable in TEGDME–based electrolyte and delivers low capacities of $\approx$20 mAh $g^{-1}$ in PC–based electrolytes, mainly due to the low oxidation resistance of ether solvent and the sluggish $Ca^{2+}$ diffusion into the host lattices in carbonate-based electrolyte, respectively (Supplementary Fig. 24). However, according to galvanostatic intermittent titration technique (GITT) (Supplementary Figs. 23b and 25) and electrochemical impedance spectroscopy (EIS) (Supplementary Fig. 26) results, the $Ca_{0.4}MnO_2$ cathode exhibits fast $Ca^{2+}$ diffusion kinetics. Such a fast kinetics is attributed to the reversible pre-insertion of protons and water molecules into the lattice of cathode material, which efficiently decreases the $Ca^{2+}$ diffusion barrier and thus improves the kinetics of batteries (Supplementary Fig. 27)[48,50].

**Electrochemical performances of the aqueous calcium-ion/sulfur batteries.** To electrochemically evaluate the ACSBs, CR2032 coin cells were assembled with a configuration of the S/C anode, $Ca_{0.4}MnO_2$ cathode, and aqueous gel electrolyte. Figure 5a shows the CV curves of the ACSB at the initial cycles,

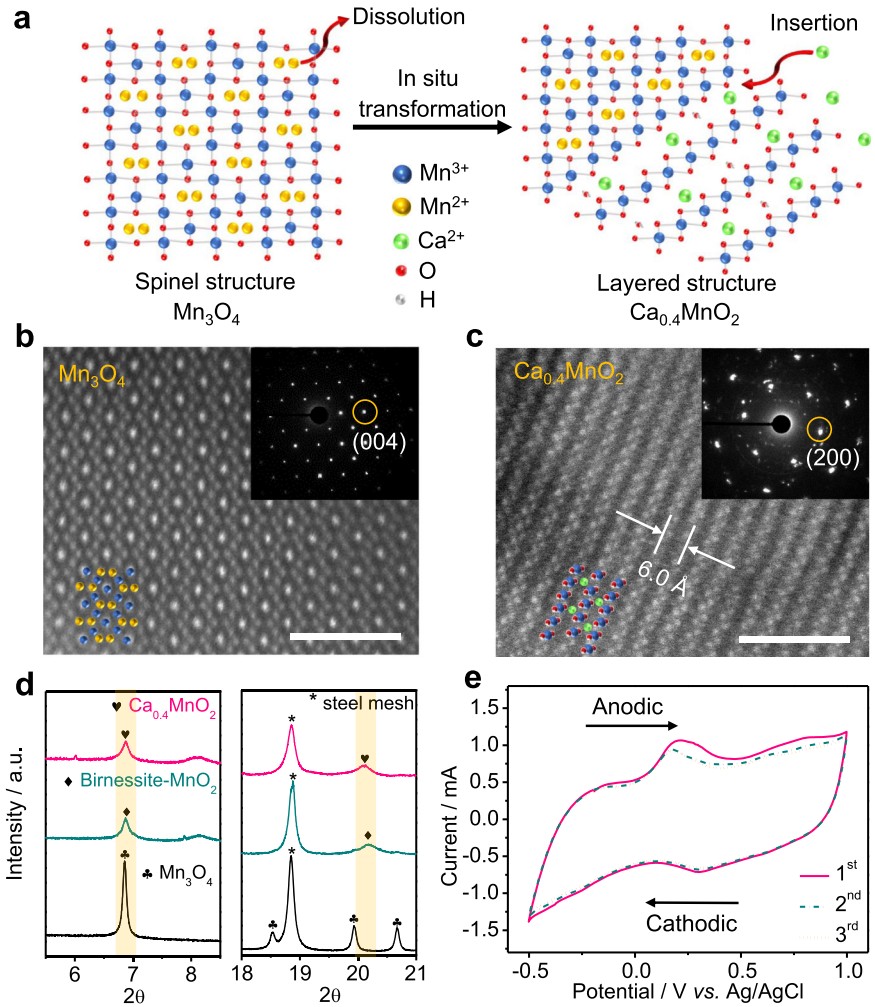

**Fig. 4 Characterization of the Ca$_{0.4}$MnO$_2$ cathode. a** Schematic illustration of the atomic structure change during the in situ electrochemical conversion. The HADDF–STEM images of **b** the Mn$_3$O$_4$ precursor and **c** The Ca$_{0.4}$MnO$_2$ cathode material. Blue, orange, green, and red balls represent Mn$^{3+}$, Mn$^{2+}$, Ca, and O atoms, respectively. Scale bars are 2 nm in Fig. 4b and c. **d** Synchrotron powder diffraction patterns of the Mn$_3$O$_4$, birnessite MnO$_2$, and Ca$_{0.4}$MnO$_2$. The peaks marked with star symbol correspond to the stainless-steel mesh current collector. **e** CV curves of the Ca$_{0.4}$MnO$_2$ cathode collected at a scan rate of 0.3 mV s$^{-1}$.

in which a pair of redox peaks are mainly attributed to the reaction as below:

$$5\,Ca_{0.4}MnO_2 + 2\,S = 2\,CaS + 5\,MnO_2 \qquad (1)$$

The 1$^{st}$ CV profile shows relatively low repeatability with the following cycles due to an activation process related to the interfacial wettability improvement and SEI formation, etc. This is consistent with the pre-activated charge/discharge profile (Supplementary Fig. 28). The subsequent CV curves are overlapped, suggesting that the cathode and anode are highly reversible in the highly concentrated aqueous gel electrolyte. Figure 5b and Supplementary Fig. 29 show the rate performance of the full cells with different electrolytes. We found that the cell cannot be stably cycled in 1 m Ca(NO$_3$)$_2$ aqueous electrolyte due to the severe HER (Supplementary Fig. 30), and shows low capacity with poor reversibility in organic-based electrolytes due to the inferior compatibility between both electrodes with such electrolytes (Supplementary Fig. 31). However, by using the aqueous gel electrolyte, the full ACSB delivers capacities of 86, 66, 46, and 35 mAh g$^{-1}$ based on the mass of total electrodes (560, 431, 302, and 228 mAh g$^{-1}$ based on the mass of sulfur) at specific currents of 0.1 C, 0.2 C,

0.5 C, and 1 C, respectively (Fig. 5c). These are higher than those using saturated Ca(NO$_3$)$_2$ aqueous electrolyte (81, 58, 36, and 25 mAh g$^{-1}$ based on the mass of total electrodes; 528, 380, 237, and 162 mAh g$^{-1}$ based on the mass of sulfur. Supplementary Fig. 32). This result indicates that the high salt concentration together with the addition of PVA not only enable the expanded electrochemical window to support the S/C‖Ca$_{0.4}$MnO$_2$ electrochemical redox chemistry, but also efficiently suppress the shuttling of calcium polysulfides. Figure 5d and Supplementary Table 2 compare the specific energies of various aqueous batteries. The S/C | gel electrolyte| Ca$_{0.4}$MnO$_2$ ACSB can achieve a high-specific energy of 110 Wh kg$^{-1}$ (based on the total mass of S/C and Ca$_{0.4}$MnO$_2$ materials as well as an average discharge voltage of ≈1.29 V, Fig. 5c. The average voltage is defined as the mid–value voltage of the discharge plateau). The specific energy achieved for ACSB is obviously higher than the previously reported counterparts, including Ca–[21], Li–[51], Na–[52–54], K–[1], Mg–[55,56], and Al–[57] based aqueous batteries. The cycling performances and corresponding Coulombic efficiencies of the ACSBs are presented in Fig. 5e. The cell with aqueous gel electrolyte exhibits superior cycling stability with 83% capacity retention after 150 cycles at 0.2 C with an average Coulombic efficiency of

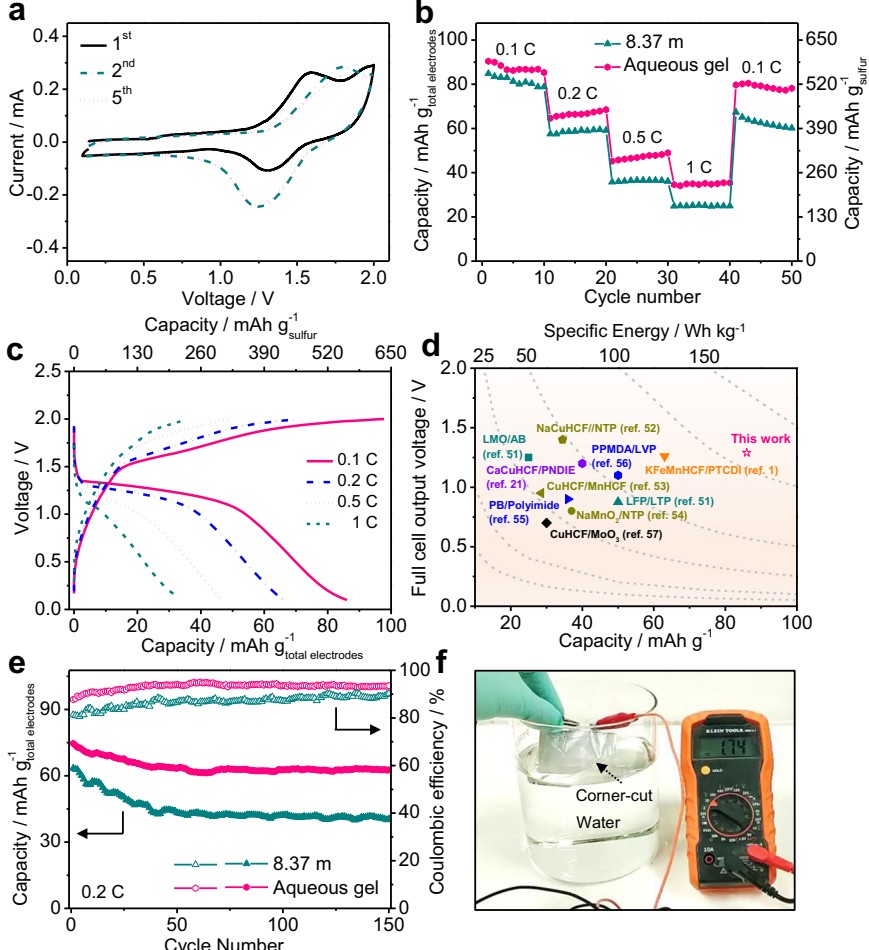

**Fig. 5 The electrochemical performances of the as–assembled ACSBs. a** The CV curves of the S/C|aqueous gel electrolyte|Ca$_{0.4}$MnO$_2$ full cell collected at a scan rate of 0.2 mV s$^{-1}$. **b** The rate performance of full cells with different electrolytes. **c** The voltage profiles of full cell assembled with aqueous gel electrolyte at different specific currents. **d** Comparison of specific energies of various aqueous energy storage devices based on the total mass of the electrode active materials. Color code: cyan, brown, orange, blue, black, and violet represent Li–, Na–, K–, Mg–, Al–, and Ca–based aqueous batteries, respectively. **e** The cycling performances and corresponding Coulombic efficiencies of S/C||Ca$_{0.4}$MnO$_2$ full cells at 0.2 C. The mass ratio of Ca$_{0.4}$MnO$_2$ cathode to S/C anode is about 1.6:1. **f** Water–soaking test of charged S/C|aqueous gel electrolyte|Ca$_{0.4}$MnO$_2$ pouch cell after corner cut.

93%. These values are much higher than those of the cell with saturated Ca(NO$_3$)$_2$ aqueous electrolyte (65 and 86%) due to the immobilization of calcium polysulfides by PVA. Single-layer S/C | aqueous gel electrolyte|Ca$_{0.4}$MnO$_2$ pouch cells were also assembled to demonstrate the high safety of the ACSBs (Supplementary Fig. 33). As shown from Fig. 5f and Supplementary Video 4, the fully charged pouch cell did not exhibit short–circuit or burning after corner cut, and even deliver a constant voltage after immersing the cut cell into water. This intrinsic safety could endow the ACSBs promising for the applications in extreme conditions, e.g., aerospace, deep-sea submarine, and other military devices.

In spite of these good electrochemical performances, it should be noted that there is still room to improve the present ACSB chemistry. Even if the Ca$_{0.4}$MnO$_2$ cathode material applied in our study seems an optimal choice, it actually has not made full use of the wide electrochemical stability window of the aqueous gel electrolyte. Thereby, the advances in exploring new cathode materials with higher capacity and redox potential could further improve the specific energy of current ACSBs. Furthermore, the Coulombic efficiency of aqueous full cells still remains relatively limited, which is possibly due to the breakdown/reconstruction of SEI and other complicated side reactions. Exploring next-

generation aqueous electrolytes with enhanced electrode compatibility to further increase the Coulombic efficiency of ACSBs remains a great challenge.

The highly concentrated aqueous gel electrolytes are universal for all aqueous multivalent–ion/sulfur chemistry, and provide a new opportunity to develop high energy, safe, and low-cost batteries. The generality of the aqueous gel electrolyte for multivalent–ion/sulfur chemistry has been further demonstrated by rechargeable aqueous Mg–ion/sulfur||metal oxide and Al–ion/sulfur||metal oxide batteries (Supplementary Figs. 34–36). The aqueous gel electrolytes prepared by dissolving 10 wt% PVA in saturated Mg(NO$_3$)$_2$ aqueous solution (Supplementary Fig. 34a,b) or in saturated Al$_2$(SO$_4$)$_3$ aqueous solution (Supplementary Fig. 35a, b) exhibit extended electrolyte stability windows (more than 2.2 V). The M$_x$MnO$_2$ (M = Mg or Al) cathodes synthesized via in situ electrochemical transformation from Mn$_3$O$_4$ show stable multivalent ion intercalation/deintercalation (Supplementary Figs. 34c and 35c), meanwhile the S/C anodes present highly reversible multivalent ion polysulfide conversion in such aqueous gel electrolytes (Supplementary Figs. 34f and 35f). The aqueous Mg–ion/sulfur and Al–ion/sulfur full cells deliver satisfactory stability during cycling (Supplementary Figs. 34d and 35d), and stable electrolyte|electrode interfaces with small resistances

(Supplementary Figs. 34e and 35e). It should be mentioned that even in the organic-based electrolytes, Ca/Mg/Al–ion batteries still suffer from either low reversibility/dendrite growth on metal anodes or low voltage/poor intercalation cathodes. We migrated these challenges by using non−flammable and cheap aqueous electrolytes, which boost the aqueous multivalent–ion batteries for low−cost large-scale energy storage.

In summary, we demonstrate that a versatile aqueous multivalent–ion/sulfur chemistry can endow the multivalent–ion batteries with high-specific energy, reversibility, and safety. By virtue of this chemistry, we developed a room-temperature aqueous Ca–ion/sulfur full battery, which was assembled with a sulfur/carbon anode, a $Ca_{0.4}MnO_2$ cathode, and an aqueous gel electrolyte. The synergistic contribution of high $Ca(NO_3)_2$ salt concentration and PVA strongly suppresses the water activity in the gel electrolyte, thus providing an expanded electrochemical stability window to support the sulfur–calcium polysulfide conversion chemistry. The sulfur/carbon anode withstands inhibited polysulfides' shuttling owing to the negligible insolubility of polysulfides in the aqueous gel electrolyte and the protection of SEI layer. Meanwhile, the in situ electrochemically synthesized $Ca_{0.4}MnO_2$ cathode exhibits highly stable $Ca^{2+}$ intercalation/deintercalation. The as-developed full calcium-ion/sulfur battery achieved a high-specific energy of 110 Wh kg$^{-1}$ with stable cyclability and excellent safety under abuse conditions. Furthermore, this battery design strategy has been successfully extended to rechargeable aqueous Mg and Al-based battery systems. These key findings make a revolutionary step–forward towards the development of aqueous multivalent–ion batteries for low-cost energy storage.

## Methods

**Materials**. The 1 m, 2 m, 5 m, and 8.37 m saturated aqueous electrolytes were prepared by dissolving $Ca(NO_3)_2$ (≈99%, Sigma–Aldrich) in deionized water after nitrogen bubbling, respectively. The gel electrolyte was prepared by dissolving 10 wt% PVA (Mw = 50000, Sigma–Aldrich) in the saturated $Ca(NO_3)_2$ aqueous electrolyte at 80 ºC under vigorous stirring.

To prepare the S/C composite, sterculia lychnophora was soaked in distilled water for 1 h followed by a freeze-dry process. The sample was ground to powders and then mixed with $KHCO_3$ in a mass ratio of 1:4. The biomass-derived porous carbon can be prepared by calcining the mixed powders at 800 °C for 5 h under argon atmosphere followed by repeatedly washing with 1 M HCl and distilled water and drying at 80 °C overnight. Then, the S/C material could be obtained by a melt–diffusion strategy, during which a mixture of porous carbon and sulfur powder (99.99%, Sigma–Aldrich) at a weight ratio of 6:4 was heated at 155 °C for 12 h. The S/C anodes were fabricated by compressing a mixture of the as-prepared S/C composite and poly(vinylidenedifluoride) (PTFE, 5 wt% aqueous solution) binder in a weight ratio of 9:1 on a titanium mesh (200 mesh) followed by vacuum drying at 60 °C. The thickness of the titanium mesh was 0.3 mm. The areal loading of sulfur was around 1–2 mg cm$^{-2}$ while the electrode area is around 0.385 cm$^2$.

The cathode precursor, $Mn_3O_4$, was synthesized as follow. Typically, 2.25 g $MnSO_4$ (≈99%, Sigma–Aldrich) was dissolved in 150 mL deionized water. Liquid ammonium hydroxide was then dropwise added into the solution under magnetic stirring until the pH reached 11. After reacting overnight at room temperature, brown $Mn_3O_4$ powders were obtained by centrifuging and washing the precipitation. After that, $Mn_3O_4$, carbon black, and PTFE (5 wt% aqueous solution) were mixed with a weight ratio of 8:1:1, and then pressed onto a stainless-steel mesh (200 mesh). The thickness of the stainless-steel mesh was 0.2 mm. The areal loading of $Mn_3O_4$ was around 7–9 mg cm$^{-2}$ while the electrode area was around 0.385 cm$^2$. Subsequently, electrochemical cells were assembled in a glass cell with the as-prepared $Mn_3O_4$ electrode as working electrode, a saturated Ag/AgCl as reference electrode, a Pt wire as counter electrode, and 5 mL 8.37 m $Ca(NO_3)_2$ aqueous electrolyte. The $Ca_{0.4}MnO_2$ electrode was electrochemically prepared by charging–discharging the electrochemical cell within a voltage window of −0.5 to 1.0 V vs. Ag/AgCl at a specific current of 50 mA g$^{-1}$ for 10 cycles.

**Characterizations**. Raman spectra were recorded via a Renishaw inVia Raman spectrometer system (Gloucestershire, UK). Nuclear magnetic resonance (NMR) and solid-state NMR measurements were performed on an Agilent 500 MHz Nuclear Magnetic Resonance and Bruker Avance III Nuclear Magnetic Resonance system, respectively. To visually observe the diffusion of calcium polysulfides in electrolyte, 1 m $CaS_4$ solution was prepared by heating a aqueous suspension of CaO and sulfur powder in stoichiometric ratio at 90 ºC under $N_2$ bubbling until all

the precipitate was dissolved. XPS measurements with depth profiles were conducted on a PHI 5000 VersaProbe II. The thickness values of the depth profiles were estimated from the $SiO_2$ calibrated sputtering. UV–vis spectra were measured on Agilent Cary 60 UV–Vis Spectrometer. Thermogravimetric analysis (TGA) was performed on a SDT2960 system under a $N_2$/Ar flow with a rate of 10 ºC min$^{-1}$. The pore volume was measured via the Brunauer–Emmett–Teller (BET) method by Micromeritics 3Flex analyzer at 77 K with $N_2$ as analysis gas. The phase information was collected by synchrotron X-ray powder diffraction with a Cu Kα X-ray source at a scan rate of 1º min$^{-1}$ (operating voltage of 40 kV and current of 25 mA). The FTIR were collected by using Nicolet Magna 6700 FTIR spectrometer. The molecular formulas were determined by ICP–OES (PerkinElmer Optima, PerkinElmer Avio). The morphologies of the samples were probed by using field emission scanning electron microscope (FE-SEM, Zeiss Supra 55VP), and HAADF–STEM (JEOL JEM–ARM200) operated at an accelerating voltage of 200 kV.

**Electrochemical measurements**. The ionic conductivities of the electrolyte samples were measured by using conductivity meter (DJS–1 C, Shanghai Leici Co., Ltd., China, in which the distance between platinum flat electrodes is 0.979 cm). During the measurement, the electrode was inserted into the electrolyte solution at room temperature (25 °C), and then the data was directly recorded. The electrochemical stability window of electrolytes was evaluated by LSV via three-electrode devices assembled with stainless-steel mesh as working electrode, saturated silver/silver chloride (Ag/AgCl) as reference electrode, and Pt wire as counter electrode at scan rate of 1 mV s$^{-1}$ (Supplementary Fig. 37). The area of the stainless-steel mesh was around 1 cm$^2$. Electrochemical cells were assembled in a glass cell via three-electrode configurations with either S/C or $Ca_{0.4}MnO_2$ as working electrode, a saturated Ag/AgCl as reference electrode, and Pt wire as counter electrode. For galvanostatic intermittent titration technique (GITT) measurements, the electrodes were charged/discharged for 10 min at a specific current of 50 mA g$^{-1}$, followed by a duration of 1 h relaxation to achieve equilibrium potential in such three-electrode device. All above electrochemical measures were conducted on a VMP3 multichannel electrochemical station (Bio Logic Science Instruments, France).

For the full cell tests, CR2032 coin cells were assembled with S/C anode, $Ca_{0.4}MnO_2$ cathode, and gel electrolyte absorbed in two pieces of glass fiber membranes (Whatman GF/A with a thickness of 260 μm each) as separators. The weight ratio of $Ca_{0.4}MnO_2$ to S/C was around 1.6:1. The electrode area of S/C anode and $Ca_{0.4}MnO_2$ cathode was around 0.385 cm$^2$, meanwhile the area of aqueous gel electrolyte was about 2.5 cm$^2$. The mass of the gel electrolyte is around 2–3 mg per milligram of total electrodes. Galvanostatic charge–discharge measurements were performed on a Neware battery test system at room temperature. Before the cycling and rate tests, the as-developed full cells were first activated by cycling at 0.1–2 V at 0.5 C for 1 cycle (1 C = 1675 mA g$^{-1}$ based on the mass of sulfur) at room temperature. CV of the assembled cells were tested using the VMP3 electrochemical workstation at a scanning rate of 0.2 mV s$^{-1}$. EISs of cells in the half-charged state were examined using the VMP3 multichannel electrochemical station in a frequency range of $10^{-2}$ to $10^5$ Hz by applying a disturbance amplitude of 5 mV. The rest time was set as 0.5 h before carrying out the EIS measurements.

## Data availability

The data that support the findings of this study are available from the corresponding author upon reasonable request.

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

## Acknowledgements

We would like to acknowledge the support by Rail Manufactory CRC projects (RMCRC: R1.1.1 and R1.1.2), and the Australian Research Council (ARC) Discovery Projects (DP200101249 and DP210101389).

## Author contributions

C.W. and G.W. conceived and designed this work. D.Z. and X.T. performed the experiments and wrote the manuscript. B.Z. conducted the molecular dynamics simulations. P.L. performed the density functional theoretical calculations. S.W., H.L., X.G., P.J., X.G., Y.F., and C.W. discussed the results and participated in the preparation of the paper.

## Competing interests

The authors declare no competing interests.
