## [Peer Review File · Nature Communications]

REVIEWER COMMENTS

Reviewer #1 (Remarks to the Author):

The authors report novel aqueous multivalent batteries based on a sulfur anode, metal oxide cathodes, and aqueous gel electrolytes by combining multiple concepts that have been introduced in recent years. Although the reversibility of e.g. the Ca-ion full cell is still quite limited with a Coulombic efficiency of 93% at C/2, the results represent significant progress in the field of multivalent aqueous batteries. The manuscript could be considered for publication after the following comments have been addressed:

- 1) In general, I think the manuscript would benefit from a more balanced discussion between the progress made and the remaining challenges. The Coulombic efficiency of the full cells remains very limited. This dramatically affects the cycling stability of practical cells with a low amount of electrolyte.
- 2) Besides capacity, Coulombic efficiency should also be plotted in Fig. 5b and 5e.
- 3) More experimental details are needed to reproduce the results:
 - a. The diameter of the SS working electrodes used for the determination of the electrochemical stability windows should be stated.
 - b. How was the stability window determined from the voltammograms? The ESW appears to be quite generously determined considering the limited Coulombic efficiency of the cell shown in Fig. 5e. The ESW should be discussed in a more balanced way.
 - c. Comparison with other publications would be facilitated by plotting current density (current divided by the geometric area of the SS electrode) instead of the absolute current in Fig. 1a.
 - d. The composition of the full cells should be described in more detail: diameters of the electrodes, amount of electrolyte (e.g. weight), diameter of the polymer electrolyte film. Was a spacer used to separate the electrodes or is the polymer electrolyte mechanically stable enough to prevent a short circuit?
- 4) The quality of the figures in terms of resolution etc. should be improved.

Reviewer #2 (Remarks to the Author):

The manuscript describes the weaknesses of the materials for multivalent aqueous battery batteries and the idea of doing this is to create the context to show another interesting system using sulfur as one of the battery materials that could be called conversion, not intercalation ones. The main contribution that the authors propose is a calcium battery using sulfur as the material of the negative electrode and that this system is also valid for other multivalent cations such as magnesium and aluminum. As electrolyte, the authors propose super concentrated $\text{Ca}(\text{NO}_3)_2$. The proposed system is a very good idea and all the concepts involved in the development of the idea are very well explored. I suggest to the authors to include some comments on their work

- 1- Which is the difference between a super concentrate electrolyte and a water-in-salt electrolyte (WiSE)?
- 2- A comparison between their battery configuration (energy, power, durability) and aqueous multivalent cations batteries.

Roberto M. Torresi

Reviewer #3 (Remarks to the Author):

In this paper, the authors report on the use of highly concentrated aqueous electrolytes in concert with sulfur anodes and high voltage intercalation cathodes to enable high-energy multivalent-ion batteries. The configuration was demonstrated with each Ca^{2+} , Mg^{2+} , and Al^{3+} as the metallic species. This work is indeed novel, and I believe many researchers in the multivalent battery

community will find this study very intriguing. The introduction, methods, results, and discussion are all quite clear. It is my opinion that this work should be published in Nature Communications.

I recommend that the manuscript be revised in response to the following minor comments before publication:

1. Line 64 - Along with sluggish cation diffusion, conversion reactions (in lieu of intercalation) have also plagued multivalent cathodes.
2. Polysulfide dissolution is investigated here. It is also known that elemental sulfur dissolution is a challenge in multivalent systems. What is the solubility of elemental sulfur in the electrolyte? Perhaps this can be measured with UV-Vis spectroscopy.
3. It is noted that trace $\text{Ca}(\text{HS})_2$ is detected by XPS. Please comment further. Is $\text{Ca}(\text{HS})_2$ formation irreversible (i.e. lead to capacity fade)?
4. Capacities are reported based on the mass of the full electrodes. This is great, but for the convenience of the readers, can you please also report on the sulfur utilization or specific capacity based on sulfur?
5. Please provide the thickness of the stainless steel mesh and titanium mesh.
6. Please elaborate for the readers on why the mesh current collectors were used in lieu of traditional flat current collectors. Why this cell design?
7. It is unclear why the ionic conductivity measurement was performed with two stainless steel mesh electrodes rather than flat blocking electrodes. It is unclear from the procedure described how the ionic conductivity of the electrolyte could be accurately extracted. Please perform this measurement with standard, flat electrodes separated by the gel electrolyte. Use of glass fiber or plastic donut are standard methods for maintaining flat electrode separation and uniform interelectrode distance with viscous or gel electrolytes.

Response letter

Reviewer #1:

The authors report novel aqueous multivalent batteries based on a sulfur anode, metal oxide cathodes, and aqueous gel electrolytes by combining multiple concepts that have been introduced in recent years. Although the reversibility of e.g. the Ca-ion full cell is still quite limited with a Coulombic efficiency of 93% at C/2, the results represent significant progress in the field of multivalent aqueous batteries. The manuscript could be considered for publication after the following comments have been addressed:

1) In general, I think the manuscript would benefit from a more balanced discussion between the progress made and the remaining challenges. The Coulombic efficiency of the full cells remains very limited. This dramatically affects the cycling stability of practical cells with a low amount of electrolyte.

Response: We thank the reviewer for the valuable comments that help to improve the quality of our work. As indicated by the reviewer, the Coulombic efficiency of the full cells remains relatively limited ($\approx 93\%$ at 0.2 C). Actually, this phenomenon is widely observed in many aqueous batteries¹. This is possibly due to the breakdown/reconstruction of solid electrolyte interphase (SEI) and other complicated side reactions^{1,2}. Thereby, exploring next-generation aqueous electrolytes with enhanced electrode compatibility to further increase the Coulombic efficiency of aqueous Ca ion-sulfur battery (ACSBs) remains a great challenge. We have added above discussion on the remaining challenge and perspective in the page 22 of the revised manuscript.

References:

- 1 Suo, L. *et al. Science* **350**, 938–943 (2015).
- 2 Jia, H. *et al. Nano Energy* **70**, 104523 (2020).

2) Besides capacity, Coulombic efficiency should also be plotted in Fig. 5b and 5e.

Response: According to the reviewers' suggestion, we have added the Coulombic efficiencies of the full cells at 0.2 C in Fig. 5e and Coulombic efficiencies of full cells at different rates in Supplementary Fig. 28, respectively.

Figure 5e The cycling performances and corresponding Coulombic efficiencies of S@C||Ca_{0.4}MnO₂ full cells at 0.2 C.

Supplementary Figure 28. Coulombic efficiencies of S@C||Ca_{0.4}MnO₂ full cells at different rates.

3) *More experimental details are needed to reproduce the results:*

a. The diameter of the SS working electrodes used for the determination of the electrochemical stability windows should be stated.

Response: In this work, the area of the stainless-steel (SS) mesh used for the determination of the electrochemical stability windows was around 1 cm². We have added this in the Methods section of the revised manuscript.

b. How was the stability window determined from the voltammograms? The ESW appears to be quite generously determined considering the limited Coulombic efficiency of the cell shown in Fig. 5e. The ESW should be discussed in a more balanced way.

Response: Thanks for your value comment. In the previously reported aqueous batteries, the electrochemical stability windows of aqueous electrolytes are mainly determined by Cyclic voltammetry (CV) or linear sweep voltammetry (LSV)¹. In this work, the electrochemical stability window of the gel electrolyte is ≈ 2.6 V at a scan rate of 1 mV s^{-1} . When decreasing the scan rate to 0.5 mV s^{-1} , the electrochemical stability window still remains ≈ 2.6 V, which can support the redox reaction of $\text{Ca}_{0.4}\text{MnO}_2$ cathode and S@C anode in the full cell (Fig. R1). Considering the phenomenon of low Coulombic efficiency widely exists in aqueous batteries where the voltage range of the redox couples is within the electrolyte stability window²⁻⁴, we speculate that the limited Coulombic efficiency of aqueous full cells could be mainly caused by breakdown/reconstruction of SEI and other complicated side reactions rather than water decomposition^{4,5}. Thereby, exploring next-generation aqueous electrolytes with enhanced electrode compatibility to further increase the Coulombic efficiency of current ACSBs remains a future research task.

Figure R1. Linear voltammetry curves of aqueous gel electrolyte recorded at 0.5 mV s^{-1} and 1 mV s^{-1} .

Reference:

- 1 Jiang, L. *et al. Nat. Energy* **4**, 495–503(2019).
- 2 Leonard, D.P. *et al. ACS Energy Lett.* **3**, 373–374 (2018).
- 3 He, X. *et al. Nat. Commun.* **9**, 1–8 (2018).
- 4 Suo, L. *et al. Science* **350**, 938–943 (2015).
- 5 Jia, H. *et al. Nano Energy* **70**, 104523 (2020).

c. Comparison with other publications would be facilitated by plotting current density (current divided by the geometric area of the SS electrode) instead of the absolute current in Fig. 1a.

Response: We appreciate the valuable comments from the reviewer. We have re-plotted linear sweep voltammetry curves with areal current density as the function of potential vs. Ag/AgCl in the revised Fig. 1a.

Figure 1a Linear voltammetry curves recorded at 1 mV s^{-1} in 1 m, 2 m, 5 m, saturated (8.37 m) $\text{Ca}(\text{NO}_3)_2$ electrolytes and aqueous gel electrolyte. The insets are the magnified views of the regions marked near anodic and cathodic extremes.

d. The composition of the full cells should be described in more detail: diameters of the electrodes, amount of electrolyte (e.g. weight), diameter of the polymer electrolyte film. Was a spacer used to separate the electrodes or is the polymer electrolyte mechanically stable enough to prevent a short circuit?

Response: We have provided these experimental details in the Methods section of the revised manuscript. For the full cell tests, CR2032 coin cells were assembled with S@C anode, $\text{Ca}_{0.4}\text{MnO}_2$ cathode, and gel electrolyte absorbed in two piece of glass fiber membranes (Whatman GF/A with a thickness of $260 \mu\text{m}$ each) as separators. The electrode area of S@C anode and $\text{Ca}_{0.4}\text{MnO}_2$ cathode was around 0.385 cm^2 , meanwhile the area of aqueous gel electrolyte was about 2.5 cm^2 . The mass of the gel electrolyte was around 2~3 mg per milligram of total electrodes.

4) The quality of the figures in terms of resolution etc. should be improved.

Response: Thanks for your valuable comment. We have updated the figures with improved resolution quality.

Reviewer #2:

The manuscript describes the weaknesses of the materials for multivalent aqueous battery batteries and the idea of doing this is to create the context to show another interesting system using sulfur as one of the battery materials that could be called conversion, not intercalation ones. The main contribution that the authors propose is a calcium battery using sulfur as the material of the negative electrode and that this system is also valid for other multivalent cations such as magnesium and aluminum. As electrolyte, the authors propose super concentrated $\text{Ca}(\text{NO}_3)_2$. The proposed system is a very good idea and all the concepts involved in the development of the idea are very well explored. I suggest to the authors to include some comments on their work

1) Which is the difference between a super concentrate electrolyte and a water-in-salt electrolyte (WiSE)?

Response: Thanks for your positive comments on the quality of our work. Generally, electrolytes can be classified into three distinct regimes based on the difference in ion solvation sheath¹. (1) “salt-in-solvent” electrolytes, where the amount of solvent molecules is higher than needed to complete the primary solvation sheath for the cations; (2) “salt-solvate” electrolytes, where the amount of solvent molecules is just sufficient to complete the primary solvation sheath for the cations, so that stoichiometric solvates often form for the largely dissociating salts. (3) “solvent-in-salt” electrolytes, where the primary solvation sheath for the cation cannot be completed due to insufficient solvent. Super concentrated electrolytes are covered by the latter two categories while the water-in-salt electrolyte belongs to the “solvent-in-salt” category.

Reference:

1 Borodin, O. *et al. Joule*, **4**, 69–100 (2020).

2) A comparison between their battery configuration (energy, power, durability) and aqueous multivalent cations batteries.

Response: We have taken the reviewers’ suggestion and provided a table to compare our work with previously reported aqueous multivalent ion batteries. This table has been added in the revised Supplementary Information as Supplementary Table 2.

Supplementary Table 2. The comparison of the electrochemical performance of the aqueous battery systems.

	Energy Density (Wh kg ⁻¹)	Power Density (W kg ⁻¹)	Durability	Ref
KFeMnHCF PTCDI	80	41	87% after 500 cycles; 73% after 2000 cycles	Nat. Energy 4 , 495–503 (2019).
NaCuHCF NTP	48.3	91	97% after 100 cycles	ChemSusChem 7 , 407–411 (2014).
CuHCF MnHCF	15	693	No Capacity loss after 1000 cycles	Nat. Commun. 5 , 1–9 (2014).
NaMnO ₂ NaTi ₂ (PO ₄) ₃	30	50	75% after 500 cycles	J. Mater. Chem. A 3 , 1400–1404 (2015).
CaCuHCF PNDIE	54	48	88% after 50 cycles	Adv. Sci. 4 , 1700465 (2017).
PB Polyimide	~35	200	60% after 2000 cycles	ACS Energy Lett. 2 , 1115–1121 (2017).

PPMDA LVP	55	106	86.8% after 1000 cycles	ACS Cent. Sci. 3 , 1121–1128 (2017).
CuHCF MoO ₃	21	350	63.7% after 100 cycles	Chem. Eng. J. 373 , 580–586 (2019).
S@C Ca _{0.4} MnO ₂	110	33	83% after 150 cycles	This work

Reviewer #3 (Remarks to the Author):

In this paper, the authors report on the use of highly concentrated aqueous electrolytes in concert with sulfur anodes and high voltage intercalation cathodes to enable high-energy multivalent-ion batteries. The configuration was demonstrated with each Ca²⁺, Mg²⁺, and Al³⁺ as the metallic species. This work is indeed novel, and I believe many researchers in the multivalent battery community will find this study very intriguing. The introduction, methods, results, and discussion are all quite clear. It is my opinion that this work should be published in Nature Communications.

I recommend that the manuscript be revised in response to the following minor comments before publication:

1) Line 64 – Along with sluggish cation diffusion, conversion reactions (in lieu of intercalation) have also plagued multivalent cathodes.

Response: We thank the reviewers' comment. Beyond the strong electrostatic interactions between multivalent ions and organic solvent molecules as well as cathode host lattices that leads to sluggish cation diffusion, the conversion reactions also plague the cathode materials in multivalent ion batteries¹. We have added this statement in the page 3 of the revised manuscript.

Reference:

1 Ling, C. *et al. Chem. Mater.* **27**, 5799–5807 (2015).

2) Polysulfide dissolution is investigated here. It is also known that elemental sulfur dissolution is a challenge in multivalent systems. What is the solubility of elemental sulfur in the electrolyte? Perhaps this can be measured with UV-Vis spectroscopy.

Response: We have performed further experiments to investigate the solubility of elemental sulfur in the aqueous electrolytes. As shown in Fig. R2a, after immersing S@C electrodes into diluted (1 m) and concentrated (8.37 m) Ca(NO₃)₂ aqueous electrolytes, the color of both the electrolytes remain clear after aging for 2 days. Furthermore, Fig. R2b presents the ultraviolet-visible (UV-vis) spectra of the 1 m and 8.37 m Ca(NO₃)₂ aqueous electrolytes with/without adding S@C electrodes. In the pristine electrolytes (e.g. 1 m and 8.37 m

Ca(NO₃)₂ aqueous solutions), the peak at ≈305 nm is assigned to the Ca(NO₃)₂ salt¹. After immersing S@C electrodes, no newly formed peaks were observed in the UV–vis spectra. This result indicates that the elemental sulfur remains stable in these aqueous electrolytes, and the loss of active materials on anode side is mainly due to the dissolution of polysulfides.

Figure R2. **a** Visual observation of S@C anode immersed in 1 m and 8.37 m Ca(NO₃)₂ aqueous electrolytes after aging for 2 days. **b** The UV–vis spectra of the bare electrolytes and electrolytes after immersing with S@C anode.

Reference:

1 Hudson, P.K. *et al. J. Phys. Chem. A* **111**, 544–548 (2007).

3) *It is noted that trace Ca(HS)₂ is detected by XPS. Please comment further. Is Ca(HS)₂ formation irreversible (i.e. lead to capacity fade)?*

Response: Thanks for your valuable comment. As shown in XPS results (Supplementary Fig. 11), CaS and trace amount of Ca(HS)₂ can be detected as the final discharge products. The CaS and Ca(HS)₂ are generated from the

reductions of solid calcium polysulfide aggregates (anhydrous) and liquefied calcium polysulfide anolytes (hydrous), respectively. Noticeably, the $\text{Ca}(\text{HS})_2$ is reversible according to the equation as followed^{1,2}:

However, the $\text{Ca}(\text{HS})_2$ is prone to dissolve into the diluted electrolyte, which leads to capacity fading in full cells. Thereby, in this work, aqueous gel electrolyte with low water activity has been applied to suppress the generation and dissolution of $\text{Ca}(\text{HS})_2$ (Fig. 3b). Meanwhile, the porous carbon host in S@C anode further restrains the shuttling of the $\text{Ca}(\text{HS})_2$. This synergistical effect endows the as-developed ACSBs with improved electrochemical performance. To investigate the reversibility of the S@C anode, we performed the XPS of S@C anode after one CV scanning process. As shown in Supplementary Fig. 13, two peaks at ≈ 164.3 and ≈ 165.3 eV are assigned to S $2\text{P}_{3/2}$ and $2\text{p}_{1/2}$ ³, indicating that the majority of sulfur species have been transformed to elemental sulfur except a small amount of residual polythionate and $\text{Ca}(\text{HS})_2$. This result is consistent with the Raman spectra shown in supplementary Fig. 10. We have added this result as Supplementary Figure 13 in the revised Supplementary Information.

Supplementary Figure 13. The S 2p XPS of the S@C anode after one CV scanning process at a scan rate of 0.5 mV s^{-1} .

Reference:

- 1 Peramunage, D. *et al. Science*, **261**, 1029–1032 (1993).
- 2 Wu, X. *et al. Nanoscale*, **9**, 11004–11011 (2017).
- 3 Yang, C. *et al. J. Am. Chem. Soc.* **137**, 2215–2218 (2015).

4) Capacities are reported based on the mass of the full electrodes. This is great, but for the convenience of the readers, can you please also report on the sulfur utilization or specific capacity based on sulfur?

Response: Thanks for the comment. Fig. 5b (right Y-axis) and Fig. 5c (upper X-axis) show the specific capacities of full cells based on the mass of sulfur at different rates. The full ACSB delivers capacities of 86, 66, 46, and 35 mAh g⁻¹ based on the mass of total electrodes (*i. e.* 560, 431, 302, and 228 mAh g⁻¹ based on the mass of sulfur) at current densities of 0.1 C, 0.2 C, 0.5 C, and 1 C, respectively. These are higher than those using saturated Ca(NO₃)₂ aqueous electrolyte (81, 58, 36, and 25 mAh g⁻¹ based on the mass of total electrodes; 528, 380, 237, and 162 mAh g⁻¹ based on the mass of sulfur. Supplementary Fig. 31).

5) Please provide the thickness of the stainless-steel mesh and titanium mesh.

Response: In this work, the thicknesses of the stainless-steel mesh and titanium mesh were 0.2 mm and 0.3 mm, respectively. We have added this in the Methods section of the revised manuscript.

6) Please elaborate for the readers on why the mesh current collectors were used in lieu of traditional flat current collectors. Why this cell design?

Response: The low-cost and chemically stable meshes are widely used as current collectors in aqueous batteries^{1, 2}. This is mainly because the areal mass loadings of electrode active materials are usually high in aqueous batteries (*e.g.* 7~9 mg cm⁻² areal loading of Ca_{0.4}MnO₂ in this work). Mesh current collectors are applied to adapt such high areal mass loadings, since they can reduce the strain on the electrodes during cycling to avoid the formation of cracks, while the electrodes on flat current collectors are prone to be cracked and then delaminated³. Moreover, mesh current collectors not only provide three-dimensional charge transport path, but also possess ample mesh opening space to accommodate active materials⁴. Additionally, the mesh current collectors are much lighter than flat current collectors with same diameter, which leads to higher energy densities of the full cells. Therefore, mesh current collectors were applied in this work. We have added above explanation in the page 13 of the revised manuscript.

Reference:

- 1 Suo, L. *et al. Science* **350**, 938–943 (2015).
- 2 Yang, C. *et al. P. Natl. Acad. Sci.* **114**, 6197–6202 (2017).
- 3 Gaikwad, A. M. *et al. Adv. Mater.* **23**, 3251–3255 (2011).
- 4 Cheng, H. Y. *et al. Chem. Eng. J.* **374**, 201–210 (2019).

7) *It is unclear why the ionic conductivity measurement was performed with two stainless steel mesh electrodes rather than flat blocking electrodes. It is unclear from the procedure described how the ionic conductivity of the electrolyte could be accurately extracted. Please perform this measurement with standard, flat electrodes separated by the gel electrolyte. Use of glass fiber or plastic donut are standard methods for maintaining flat electrode separation and uniform interelectrode distance with viscous or gel electrolytes.*

Response: Thank you for the valuable comment. We re-measured the ionic conductivities of aqueous electrolytes by using conductivity meter (DJS-1C, Shanghai Leici Co., Ltd., China). The DJS-1C conductivity meter consists of two platinum flat electrodes and the distance between platinum flat electrodes is 0.979 cm. During the measurements, the electrode was inserted into the electrolyte solution at room temperature (25 °C), and then the data was directly recorded (inset in Supplementary Fig. 1). The ionic conductivity values are shown in Supplementary Fig. 1, which is similar to the previously obtained values tested via stainless steel electrodes. We have added this experimental detail in the revised manuscript and the result in the revised Supplementary Information, respectively.

Supplementary Figure 1. The electrochemical conductivities and pH values of the 1 m, 2 m, 5 m, Saturated (8.37 m) $Ca(NO_3)_2$ aqueous solutions, and gel electrolyte. The inset shows the schematic illustration of ionic conductivity testing process via conductivity meter.

REVIEWERS' COMMENTS

Reviewer #1 (Remarks to the Author):

The authors prepared a satisfactory revision including additional experiments. In my opinion, the manuscript can now be accepted for publication.

Reviewer #2 (Remarks to the Author):

Authors has answered all comments and I think that the revised manuscript is now ready to be published.

Reviewer #3 (Remarks to the Author):

I recommend publication once the reviewers have made the following final minor amendments.

- The authors responded to Reviewer #1 Question 3b regarding how the voltammograms were used to extract the reported electrochemical stabilities. However, the response is somewhat generic and figures were not added to the supporting information so that readers are able to see how the values were determined. Please state the criteria for choosing the exact oxidative and reductive limit potentials and show graphically in the supporting information.
- With regards to the response to Reviewer #3 Question 2: (1) Elemental sulfur is colorless at low concentration in solution (unlike polysulfides), therefore the photograph shown does not give us any information about elemental sulfur solubility in the electrolyte. (2) The fact that the UV-Vis data is cut off at 250 cm^{-1} makes it a bit difficult to tell. Elemental sulfur at low concentration absorbs near 250 cm^{-1} . (3) Please at least state something in the manuscript about how elemental sulfur solubility is an also important consideration for sulfur-based batteries.

Response letter

Reviewer #1:

The authors prepared a satisfactory revision including additional experiments. In my opinion, the manuscript can now be accepted for publication.

Response: We thank the reviewer for the positive comment.

Reviewer #2:

Authors has answered all comments and I think that the revised manuscript is now ready to be published.

Response: Thanks for your positive comment on the quality of our work.

Reviewer #3 (Remarks to the Author):

I recommend publication once the reviewers have made the following final minor amendments.

1) The authors responded to Reviewer #1 Question 3b regarding how the voltammograms were used to extract the reported electrochemical stabilities. However, the response is somewhat generic and figures were not added to the supporting information so that readers are able to see how the values were determined. Please state the criteria for choosing the exact oxidative and reductive limit potentials and show graphically in the supporting information.

Response: We thank the reviewers' comment. Hydrogen evolution reaction (HER)/oxygen evolution reaction (OER) will start when the corresponding current density dramatically increases during linear sweep voltammetry (LSV) test¹. In this work, the electrochemical stability window was recorded as the voltage range where the HER current density was lower than 0.5 mA cm^{-2} and OER current density was lower than 0.1 mA cm^{-2} , which is consistent with previous reports^{1,2}. When decreasing the scan rate to 0.5 mV s^{-1} , the electrochemical stability window still remains $\approx 2.6 \text{ V}$, which can support the redox reaction of $\text{Ca}_{0.4}\text{MnO}_2$ cathode and sulfur/carbon (S/C) anode in the full cell (Supplementary Fig. 37). Therefore, the scan rate of LSV was set as 1 mV s^{-1} in this work. We have added above result and explanation in the revised supplementary information as Supplementary Fig. 37.

Supplementary Figure 37. Linear voltammetry curves of aqueous gel electrolyte recorded at 0.5 mV s^{-1} and 1 mV s^{-1} .

Reference:

- 1 Jiang, L. *et al. Nat. Energy* **4**, 495–503(2019).
- 2 Suo, L. *et al. Science* **350**, 938–943 (2015).

2) *With regards to the response to Reviewer #3 Question 2: (1) Elemental sulfur is colorless at low concentration in solution (unlike polysulfides), therefore the photograph shown does not give us any information about elemental sulfur solubility in the electrolyte. (2) The fact that the UV–Vis data is cut off at 250 cm^{-1} makes it a bit difficult to tell. Elemental sulfur at low concentration absorbs near 250 cm^{-1} . (3) Please at least state something in the manuscript about how elemental sulfur solubility is an also important consideration for sulfur–based batteries.*

Response: We thank the reviewer for the valuable comments. As suggested by reviewer, we have updated the ultraviolet–visible (UV–vis) spectra with the wavelength ranging from 200 to 800 nm. As shown in the UV–vis spectra of the 1 m and 8.37 m $\text{Ca}(\text{NO}_3)_2$ aqueous electrolytes with/without adding S/C electrodes, the pristine electrolytes (*e.g.* 1 m and 8.37 m $\text{Ca}(\text{NO}_3)_2$ aqueous solutions) present a peak at $\approx 305 \text{ nm}$, which is assigned to the $\text{Ca}(\text{NO}_3)_2$ salt¹. After immersing S/C electrodes, no newly formed peaks were observed in the UV–vis spectra. This result indicates that the solubility of elemental sulfur is negligible in the aqueous electrolytes. We have added the explanation in the revised manuscript on Page 10 and UV–vis result in the revised supplementary information as Supplementary Fig. 8, respectively.

Supplementary Figure 8. The UV-vis spectra of the bare electrolytes and electrolytes after immersing with S/C anode.

Reference:

- 1 Hudson, P.K. *et al. J. Phys. Chem. A* **111**, 544–548 (2007).